# Urinary KIM-1 Correlates with the Subclinical Sequelae of Tubular Damage Persisting after the Apparent Functional Recovery from Intrinsic Acute Kidney Injury

**DOI:** 10.3390/biomedicines10051106

**Published:** 2022-05-10

**Authors:** Cristina Cuesta, Isabel Fuentes-Calvo, Sandra M. Sancho-Martinez, Floris A. Valentijn, Annette Düwel, Omar A. Hidalgo-Thomas, Consuelo Agüeros-Blanco, Adalberto Benito-Hernández, María A. Ramos-Barron, Carlos Gómez-Alamillo, Manuel Arias, Tri Q. Nguyen, Roel Goldschmeding, Carlos Martínez-Salgado, Francisco J. López-Hernández

**Affiliations:** 1Institute of Biomedical Research of Salamanca (IBSAL), 37007 Salamanca, Spain; ccuesta@usal.es (C.C.); ifc@usal.es (I.F.-C.); smsanchom@usal.es (S.M.S.-M.); annette@usal.es (A.D.); hidalgothomas@usal.es (O.A.H.-T.); 2Translational Research on Renal and Cardiovascular Diseases (TRECARD)-REDINREN (ISCIII), Departamento de Fisiología y Farmacología, Universidad de Salamanca, 37007 Salamanca, Spain; 3Department of Pathology, University Medical Centre Utrecht, 3508 GA Utrecht, The Netherlands; f.a.valentijn@umcutrecht.nl (F.A.V.); t.q.nguyen@umcutrecht.nl (T.Q.N.); r.goldschmeding@umcutrecht.nl (R.G.); 4Servicio de Nefrología, Hospital Universitario Marqués de Valdecilla, Instituto de Investigación Sanitaria Valdecilla (IDIVAL), 39011 Santander, Spain; mconsuelo.agueros@scsalud.es (C.A.-B.); adalberto.benito@scsalud.es (A.B.-H.); ramosbarron@gmail.com (M.A.R.-B.); cgalamillo@gmail.com (C.G.-A.); nefarm@gmail.com (M.A.)

**Keywords:** KIM-1, subclinical sequelae, acute kidney injury, biomarker

## Abstract

Acute kidney injury (AKI) poses an increased risk factor for new AKI episodes, progression to chronic kidney disease, and death. A worsened evolution has been linked to an incomplete renal repair beyond the apparent functional recovery based on plasma creatinine (pCr) normalization. However, structural sequelae pass largely unnoticed due to the absence of specific diagnostic tools. The urinary kidney injury molecule 1 (KIM-1) participates in renal tissue damage and repair and is proposed as a biomarker of early and subclinical AKI. Thus, we study in this paper the evolution of KIM-1 urinary excretion alongside renal tissue sequelae after an intrinsic AKI episode induced by cisplatin in Wistar rats. Creatinine clearance, pCr, proteinuria and the fractional excretion of Na^+^ and glucose were used to monitor renal function. Renal tissue damage was blindly scored in kidney specimens stained with hematoxylin–eosin and periodic acid–Schiff. KIM-1 urinary excretion and renal mRNA expression were also assessed. Finally, we analyzed urinary KIM-1 in patients apparently recovered from AKI. Our results show that, after the normalization of the standard markers of glomerular filtration and tubular function, the extent of persistent histological findings of tissue repair correlates with the renal expression and urinary level of KIM-1 in rats. In addition, KIM-1 is also elevated in the urine of a significant fraction of patients apparently recovered from an AKI. Besides its potential utility in the early and subclinical diagnosis of renal damage, this study suggests a new application of urinary KIM-1 in the non-invasive follow-up of renal repair after AKI.

## 1. Introduction

According to international consensus criteria, acute kidney injury (AKI) is defined as a sudden loss of the renal excretory function manifested as an increase in plasma creatinine (pCr) concentration or oliguria [1]. AKI is associated with high morbidity and mortality rates, especially in the context of intensive care medicine [2], and its incidence increases yearly due to population aging and attendant comorbidity (e.g., diabetes and hypertension) [3]. AKI affects 20.9% of hospital admissions and 3000–5000 (or even as high as 15,000) patients per million population annually [4], a percentage increasing to 39% in intensive care units [5].

Despite renal dysfunction being resolved in two-thirds of AKI patients within 3–7 days [6], the classical view of AKI as a transitory, reversible, and mostly inconsequential syndrome has changed significantly in the last decade. AKI increases the risk of subsequently developing chronic kidney disease (CKD) [7], increases the odds of cardiovascular morbidity, and predisposes to new AKI episodes and death. Recovery markedly determines subsequent evolution [8]. Whilst worse prognosis is linked to deficient recovery, full recovery is also associated with increased morbidity odds [9], even after mild [10,11] and subclinical AKI episodes [12,13].

The concept of recovery from AKI is still sub-optimally profiled. A consensus definition was provided by the Acute Dialysis Quality Initiative (ADQI) group [14], which relies on renal function improvement based on pCr. This definition is restricted by the intrinsic limitations of pCr as a biomarker of renal performance [3,15]. For instance, a substantial decrease in glomerular filtration rate (GFR) may be needed for pCr to increase over the normality range. In addition, due to the recruitment of the renal functional reserve increasing single nephron GFR, a significant damage involving an extensive number of nephrons is necessary for overall GFR to decrease. A corollary follows that, during AKI recovery, pCr may return to normal levels in advance to full GFR restoration. In the case of intrinsic forms of AKI, such as acute tubular injury (ATI), in which the decline in GFR is caused by tubular injury [16], pCr normalization may thus occur along with incomplete tissue repair.

Because incomplete and maladaptive repair has been linked to subsequent morbidity and progression to CKD [17], further diagnostic granularity incorporating structural restoration is required for a precise monitorization of the subclinical sequelae of ATI. Therefore, Kashani and Kellum [8] suggested to include injury biomarkers, such as kidney injury molecule 1 (KIM-1), in the definition of recovery. KIM-1 is a type-1 membrane protein whose ectodomain is released into the lumen of acutely and chronically injured proximal tubules and appears in the urine. KIM-1 is believed to exert a protective role during early and acute injury, while participating in maladaptive damage chronification [18]. Urinary KIM-1 is used as a biomarker of kidney injury in experimental models and patients, outperforming pCr as a predictor of histopathological changes caused by ischemia and drugs [19]. Interestingly, a transcriptional analysis revealed that the expression of the Havcr1 gene (i.e., coding for KIM-1) was upregulated in mouse kidneys during AKI repair [20]. Further to these grounds, the aim of the present study is to study the evolution of urinary KIM-1 during and, especially, after ATI in relation to the potential structural sequelae and repair process.

## 2. Materials and Methods

All reagents were purchased from Merck (Madrid, Spain), except where otherwise indicated.

### 2.1. In Vivo Experimental Model

We used male Wistar rats (240–260 g, 12 weeks old) in accordance with the Principles of the Declaration of Helsinki and the European Guide for the Care and Use of Laboratory Animals (Directive 2010/63/UE) and Spanish national and regional regulations (Law 32/2007/Spain, RD 1201/2005 and RD 53/2013). All experimental procedures were approved by the Bioethics Committee for Animal Care and Use of the University of Salamanca. Rats were subdivided into two experimental groups (*n* = 6 each): control group (C): rats receiving saline solution (0.9% NaCl, i.p.); cisplatin group (CP): rats receiving cisplatin (5 mg/kg body weight, i.p.). The nephrotoxic dose of cisplatin was selected from our previous studies [21]. We evaluated renal function in the following time points: basal (B), prior to cisplatin administration; day of maximum kidney damage (D4), i.e., the day in which pCr reaches the highest level; day of recovery (R0), i.e., the day in which pCr returns to basal levels; weekly after R0 for four weeks (R1, R2, R3 and R4, respectively). On these same time points, kidneys were perfused and dissected under i.p. pentobarbital anaesthesia (60 mg/kg; Vetoquinol, Alcobendas, Spain), and rats were sacrificed by exsanguination. A flow chart of the experimental protocol is shown in Figure 1.

### 2.2. Sample Collection

Urine and plasma samples were collected to evaluate renal function in the indicated time points. We collected 24 h urine in metabolic cages under controlled conditions, centrifuged and stored at −80 °C. Blood was drown from the tail vein, centrifuged and plasma was stored at −80 °C. Kidney perfusion was performed with saline solution through the aorta at the indicated time points, and the kidneys were immediately dissected and longitudinally divided in two equal halves. One half was frozen in liquid nitrogen and stored at −80 °C, and the other half was fixed in buffered 3.7% p-formaldehyde for histological analysis.

### 2.3. Renal Function Studies

Plasma (pCr) and urinary creatinine (uCr) and proteinuria were analysed using commercial kits based, respectively, on Jaffe’s reaction and the Bradford method, following the manufacturer’s instructions (BioAssay System, Hayward, CA USA). The glomerular filtration rate (GFR) was measured by the creatinine clearance (ClCr), according to the following formula:ClCr = uCr × UF/pCr, were UF is the urine flow.

Urinary and plasma sodium were determined with an automated method (LAQUATWin B-722, Horiba Scientific, Madrid, Spain). Plasma glucose was determined with the Contour^®^XT glucometer (Ascensia, Barcelona, Spain). Urinary glucose was measured using the o-toluidine method: briefly, urine was mixed with trichloroacetic acid and o-toluidine and heated at 80 °C for 20 min; when the mix was cold, absorbance (630 nm) was measured in a spectrophotometer. The fractional excretion of sodium (FENa) and glucose (FEGlc) were calculated as 100 × (urinary Na or Glc × pCr)/(plasma Na or Glc × uCr).

### 2.4. Histological Studies

Paraformaldehyde-fixed rat kidney samples were immersed in paraffin and cut into 5 μm thick slices. Haematoxylin–eosin (HE) and Periodic acid–Schiff (PAS) staining was performed. Microphotographs were obtained using DotSlide virtual microscopy technique (Olympus BX51, Olympus Iberia, Barcelona, Spain). Image analysis was performed with Olyvia Software (Olympus Iberia). The severity of renal damage was assessed blindly by a renal pathologist. Signs of tubular injury, inflammation and fibrosis were assessed on PAS- and HE-stained slides. Injury was graded on a scale from 0 to 3 as a percentage of the total cortical area of the tissue section: 0 = 0%; 1 = 1–25%; 2 = 25–50%; 3 = >50%. Signs of acute tubular injury included tubular dilatation, cast formation, enlarged nuclei, epithelial necrosis and mitosis activity, and are based on published criteria [22]. Inflammation was defined as interstitial leukocyte infiltration and fibrosis was defined as the cortical area affected by interstitial fibrosis and tubular atrophy (IFTA).

### 2.5. Patients and Clinical Protocol

Urine samples were collected from 42 volunteers from the Nephrology Department (Hospital Universitario Marqués de Valdecilla, Santander, Spain), who provided written consent. Volunteers were 30 AKI patients who had recovered from an AKI episode (i.e., their pCr had returned to the values prior to the AKI episode) and 12 controls who had not suffered an AKI. Renal function was monitored by means of serum Cr (from their medical records), and AKI was defined and classified according to Kidney Disease: Improving Global Outcomes (KDIGO) criteria (Khwaja, 2012). Urine samples were collected at the time of serum Cr normalization (or return to levels prior to the AKI episode, in the case of CKD patients) and was used to measure KIM-1 (as described below). All protocols were approved by the local Ethics Committee and were conducted according to the principles established in the Declaration of Helsinki (World Medical Assembly), the Council of Europe Convention on Human Rights and Biomedicine, the UNESCO Universal Declaration on the Human Genome and Human Rights, the requirements established in the Spanish legislation in the field of biomedical research, personal data protection and bioethics, as well as the provisions of the Law 14/2007, of 3 July, of Biomedical Research; and RD 53/2013, of 1 February.

### 2.6. Analysis of Urinary KIM-1 Excretion

Rat urinary KIM-1 was quantified with ELISA kits (Cusabio, Houston, TX, USA; urine dilution: 1/10), following the manufacturer’s instructions. Urinary concentration was corrected by the urinary flow to express data as urinary excretion. Human urinary KIM-1 was determined by Western blot. A total of 21 μL from each urine sample were separated by acrylamide electrophoresis. Proteins were transferred to an Immobilon-P Transfer Membrane (Millipore, Madrid, Spain) and incubated with a primary antibody for KIM-1 (LS-B2031, LifeSpan BioSciences, Derio, Vizcaya, Spain, dilution: 1/500 in 3% bovine serum albumin), followed by horseradish peroxidase-conjugated secondary antibody and chemiluminescent detection (Immobilon Western Chemiluminescent HRP Substrate kit; Millipore, Madrid, Spain) with the ChemiDocTMMP Imaging System (Bio-Rad, Madrid, Spain). Bands were quantified with the Scion Image software (Scion Corporation, Frederick, ML, USA), and normalized to the signal of a positive control (as percentual arbitrary units, %AU), loaded in all gels. The positive control consisted of a urine sample from a designated AKI patient with increased biomarker excretion, used as a trans normalization control in all experiments.

### 2.7. Renal Gene Expression Analysis

Quantitative RT-PCR analysis was performed in triplicate as previously described [23]. Briefly, RNA was isolated from renal tissue homogenates using the RNeasy Mini Kit (Quiagen, Madrid, Spain). For this purpose, renal tissue was pulverized, and 10–12 mg was lysed in RLT buffer supplemented with 1% β-mercaptoethanol. The lysate was diluted with 70% ethanol and filtered through columns. The RNA obtained was quantified with a Nanodrop ND-1000 type spectrophotometer (NanoDrop Technologies, Wilmington DE, USA) and stored at −80 °C. iScript RT Supermix 5X (Bio-Rad, Hercules, CA, USA) was used to generate single-stranded cDNA from 400 ng of total RNA. Each 20 μL reaction contained 1 μL of cDNA, 400 nmol/L of KIM-1 primer, and 1× iQ SybrGreen Supermix (Bio-Rad). Gene expression was normalized to ribosomal protein 7 (RPL7) expression. Thermocycling was performed on an iQ5 Real-time PCR detection system (Bio-Rad). The primers used for KIM-1 were the following: forward: 5′-GTGAGTGGACCAGGCACACA-3′ and reverse: 5′-AATCCCTTGATCCATTGTTTTCTT-3′. For RPL7: forward: 5′-CGGTCTAGACAACAAGCTGC-3′ and reverse: 5′-CACGAAGGCCCCAAAAGTG-3′. Cycling conditions: 95 °C for 5 min, 95 °C for 30 s, 65 °C for 30 s, 72 °C for 30 s (×40 cycles) and, finally, one cycle at 72 °C for 5 min.

### 2.8. Statistical Analysis

The GraphPad Prism 7 software (San Diego, CA, USA) was used for statistical analysis. Data normal distribution was assessed by the Shapiro–Wilk normality. Data are represented as the mean ± standard error of the mean (SEM). Comparison between time points were performed by a one-way ANOVA with Bonferroni’s test (for data with normal distribution) or Dunn’s test (for data with non-normal distribution). Comparison between two experimental groups on the same time point were carried out using the Student’s *t*-test. Spearman correlation studies were used to analyse the association of KIM-1 to renal damage. A value of *p* < 0.05 was considered statistically significant.

## 3. Results

### 3.1. Kidneys Show Structural Sequelae That Remain after the Normalization of the Glomerular Filtration Rate and Tubular Function

Cisplatin induces an AKI episode after approximately four days (D4), characterized by significant increases in pCr, proteinuria and in the fractional excretion of glucose and sodium, and a sharp decrease in ClCr. These changes are consistent with severe acute tubular injury (ATI), an intrinsic form of AKI characterized by tubular necrosis with an impairment in glomerular filtration. All these parameters return to baseline values about 4–6 days after D4 (R0) and remain at normal levels during the follow-up period (i.e., up to 4 weeks after R0) (Figure 2).

The severe characteristics of ATI observed four days after cisplatin injection (D4) include cast formation, enlarged nuclei, effacement of the brush border, tubular epithelial necrosis and tubular dilatation, as evidenced by haematoxylin–eosin (Figure 3) and PAS (Figure 4) stainings. These lesions remain to a variable extent after the normalization of pCr and ClCr. Some of them are directly related to the tubular damage (tubular necrosis, loss of the brush border, the luminal accumulation of cellular debris, enlarged nuclei and inflammation), whilst others are reflective of repair and remodeling (epithelial disorganization, cell proliferation and fibrosis). Tubular dilatation and enlarged nuclei were still detected three weeks after ClCr normalization (i.e., on R3), and cast formation, inflammation and fibrosis on R4 (Figure 5).

### 3.2. KIM-1 Urinary Excretion Remains Elevated Three Weeks after the Apparent Normalization of Renal Function

Both urinary KIM-1 excretion and concentration are significantly increased on the day of maximum AKI (D4), but still increase and peak by R0 (i.e., at ClCr normalization). Urinary KIM-1 levels remain significantly elevated even three weeks afterwards, by R3 (Figure 6). KIM-1 renal mRNA expression follows a similar pattern to that observed in the urinary excretion of this protein throughout the entire experimental period (Figure 6).

### 3.3. Urinary KIM-1 Excretion Correlates with the Degree of Histological Damage

A statistically significant correlation was observed between urinary KIM-1 excretion or concentration and several of the histological damage parameters analyzed. The strongest correlations were found between the urinary levels of KIM-1 and tubular epithelial necrosis, followed by tubular dilatation, cast formation and enlarged nuclei (Table 1). However, there are no associations between KIM-1 excretion and repair or remodeling proxies, such as mitosis activity, inflammation or fibrosis.

### 3.4. KIM-1 Is Elevated in the Urine of Patients Apparently Recovered from an Episode of AKI

We analyzed urinary KIM-1 levels in patients who had recovered from an AKI episode (i.e., whose serum Cr had returned to the level before the AKI episode). We distinguished between patients with previous CKD (i.e., chronic patients) from those with previous normal kidney function (i.e., non-chronic patients). We found that, upon AKI recovery, non-chronic patients have heterogeneous urinary KIM-1 levels, ranging from very high to very low or normal levels (Figure 7). This indicates that a significant number of these patients might have subclinical sequelae after the AKI episode, despite their renal function had apparently normalized. We also found that all chronic patients, except one of them, showed high levels of urinary KIM-1 after AKI recovery, in agreement with previous reports associating KIM-1 to CKD.

## 4. Discussion

This study shows that, at least during four weeks after the apparent recovery of renal function from an intrinsic AKI episode in rats (estimated by normalization of pCr), structural alterations remain. These alterations are related in part to the damage process (necrosis and tubular debris, brush border effacement and enlarged nuclei), but the alterations related to the renal repair process (epithelial disorganization, tubular dilatation, cell proliferation, fibrosis and inflammation) predominate, which gradually disappear as the kidney structure regenerates. Some of these alterations correlate with the increased urinary excretion of KIM-1. The information provided by KIM-1 in this context is distinct from and complementary to that provided by biomarkers of tubule functional performance, such as the fractional excretion of sodium and glucose. In fact, both parameters are normalized apace with pCr and ClCr, indicating that still significantly injured kidneys seem to cope with critical functions including glomerular filtration and tubule reabsorption. Overall, our study suggests that urinary KIM-1 might be reflective of subclinical histological sequelae of acute tubular injury and the regeneration processes.

KIM-1 is a glycoprotein of the T-cell immunoglobulin and mucin domain family of proteins that participates in many physiological and pathophysiological processes. In the context of kidney disease, KIM-1 is involved in adaptive reactions of the kidney epithelium to AKI and in the progression of CKD (reviewed in [24]). Ischemic and toxic damage induce KIM-1 expression on the apical surface of murine proximal tubular epithelial cells [25,26], which is interpreted as part of an adaptive response where KIM-1 becomes a player of both injury and repair [27]. Indeed, KIM-1 participates in the transformation to a phagocytic phenotype [28] and the maintenance and restoration of proximal tubular epithelial integrity [29,30], and promotes the migration and proliferation of dedifferentiated cells during renal epithelium regeneration [31].

KIM-1 expression and its plasmalemmal level increase in damaged tubule cells and, interestingly, persist until cells have completely recovered [19]. Thereupon, KIM-1 may appear in the urine due to the proteolytic cleavage of its extracellular domain and also after epithelial cell death [28]. The ectodomain of KIM-1, when shed to the urine during renal damage and repair, is also a sensitive biomarker of AKI whose excretion is associated with the severity of the underlying renal pathological processes [28]. The dual role of KIM-1 in renal tissue damage and repair is thus expected to result in a composite urinary excretion reflecting an undetermined mixture of both processes evolving during and after an AKI episode. The elevated levels of urinary KIM-1 that we found on D4 in our model are possibly due mostly to tissue damage, but the even higher levels detected after pCr normalization (R0) are more likely to be also indicative of repair and regeneration. Therefore, the information provided by the isolated measurements of urinary KIM-1 should thus be timely contextualized for etiopathological conceptualization and diagnostic application. However, the follow up of urinary KIM-1 estimated by repeated determinations through time might be indicative of pathological evolution. Whilst tapering levels would reflect resolution, persistent or increasing levels would indicate chronification and progression, respectively.

Several clinical studies have shown that urinary KIM-1 is a sensitive biomarker of cisplatin nephrotoxicity [32,33]. The urinary level of KIM-1 correlates with the severity of histological damage, which proposes this marker as a potential predictor of adverse renal outcomes after AKI [34,35]. By projecting our preclinical results onto the interpretation of the clinical scenario, our data suggest that, according to their elevated urinary KIM-1 level, many patients in our study population would bear an ongoing process of renal tissue damage or repair. This process may be the subclinical consequence of the AKI from which they had just apparently recovered, or a pre-existing chronic kidney disease. Thus, a basal (i.e., pre-AKI) measurement is necessary for a correct interpretation of urinary KIM-1 after AKI. Presently, this is only attainable in those circumstances in which the cause of AKI can be anticipated, such as before the inception of therapeutic regimes with nephrotoxic drugs or before the application of medical procedures potentially harmful for the kidneys (e.g., contrast radiographies, surgery, etc.).

To summarize, this study demonstrated the presence of structural and functional alterations after the apparent recovery from an AKI episode and highlighted the role of urinary KIM-1 as an indicator of these subclinical sequelae. A limitation to this study and an issue for prospective research is the identification of the specific pathophysiological mechanisms and trafficking pathways linking the identified structural sequelae with the increased urinary excretion of KIM-1. In perspective, our findings propose a new diagnostic application of KIM-1 for the non-invasive follow-up of subclinical renal recovery.

## Figures and Tables

**Figure 1 biomedicines-10-01106-f001:**
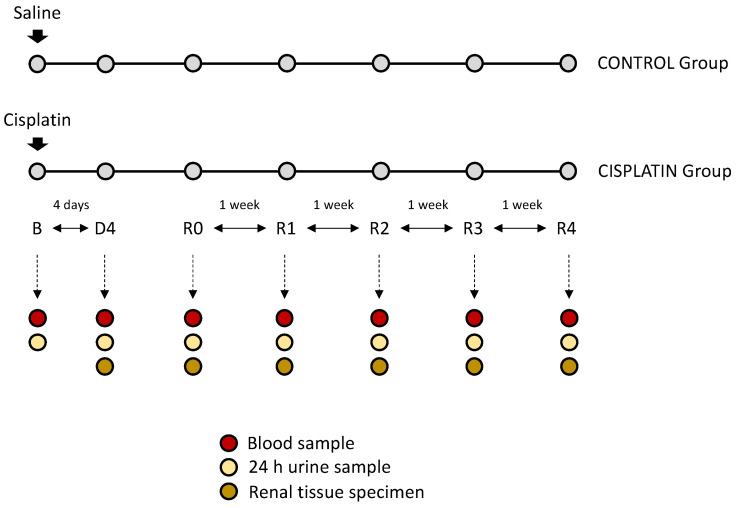
Flowchart of the study design. B: basal timepoint; D4: day of maximum kidney damage after cisplatin treatment; R0: day of recovery; R1: 1 week after recovery; R2: 2 weeks after recovery; R3: 3 weeks after recovery; R4: 4 weeks after recovery.

**Figure 2 biomedicines-10-01106-f002:**
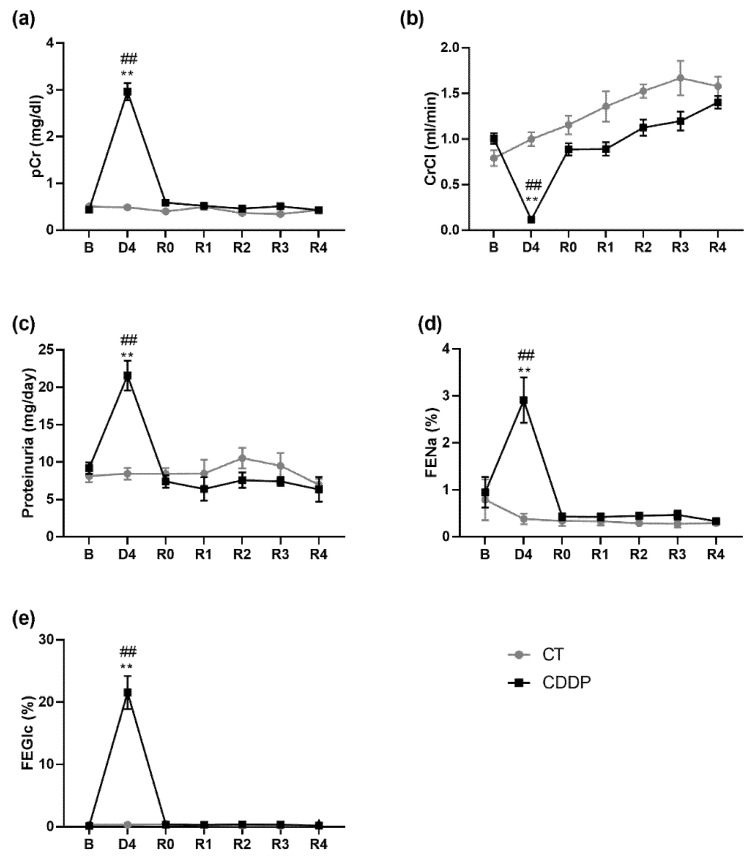
Renal function parameters: plasma creatinine (pCr) (**a**), creatinine clearance (ClCr) (**b**), proteinuria (**c**), fractional excretion of sodium (FENa) (**d**) and fractional excretion of glucose (FEGlc) (**e**). B: basa; CDDP: cisplatin treatment (5 mg·kg^−1^ body weight) group; CT: control group; D4: day of maximum kidney damage after cisplatin treatment; R0: day of recovery; R1: 1 week after recovery; R2: 2 weeks after recovery; R3: 3 weeks after recovery; R4: 4 weeks after recovery. ** *p* < 0.01 vs. B; ## *p* < 0.01 vs. CT.

**Figure 3 biomedicines-10-01106-f003:**
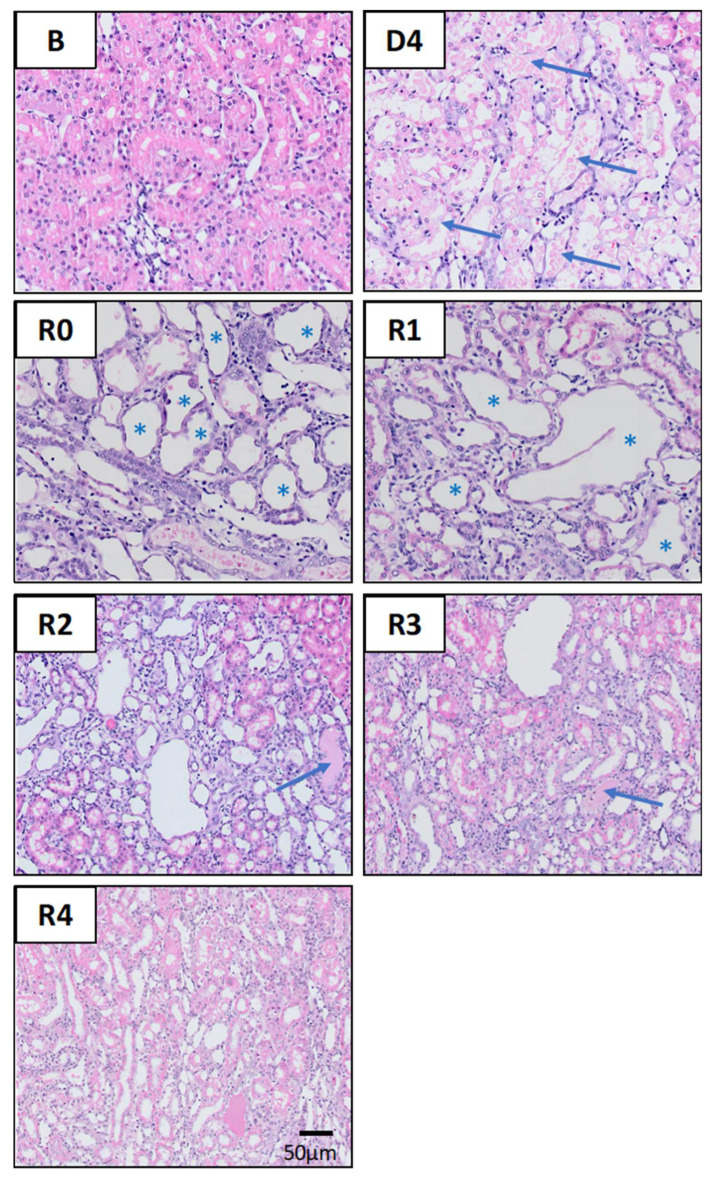
Renal histology. Representative images of kidney specimens stained with haematoxylin and eosin (HE) at different time points of AKI evolution. (**B**): basal; (**D4**): day of maximum kidney damage after cisplatin treatment; (**R0**): day of recovery; (**R1**): 1 week after recovery; (**R2**): 2 weeks after recovery; (**R3**): 3 weeks after recovery; (**R4**): 4 weeks after recovery. Arrows in D4 indicate widespread detachment of necrotic tubular epithelial cells. Asterisks in R0 and R1 indicate tubular dilatation. Arrows in R2 and R3 indicate intratubular casts.

**Figure 4 biomedicines-10-01106-f004:**
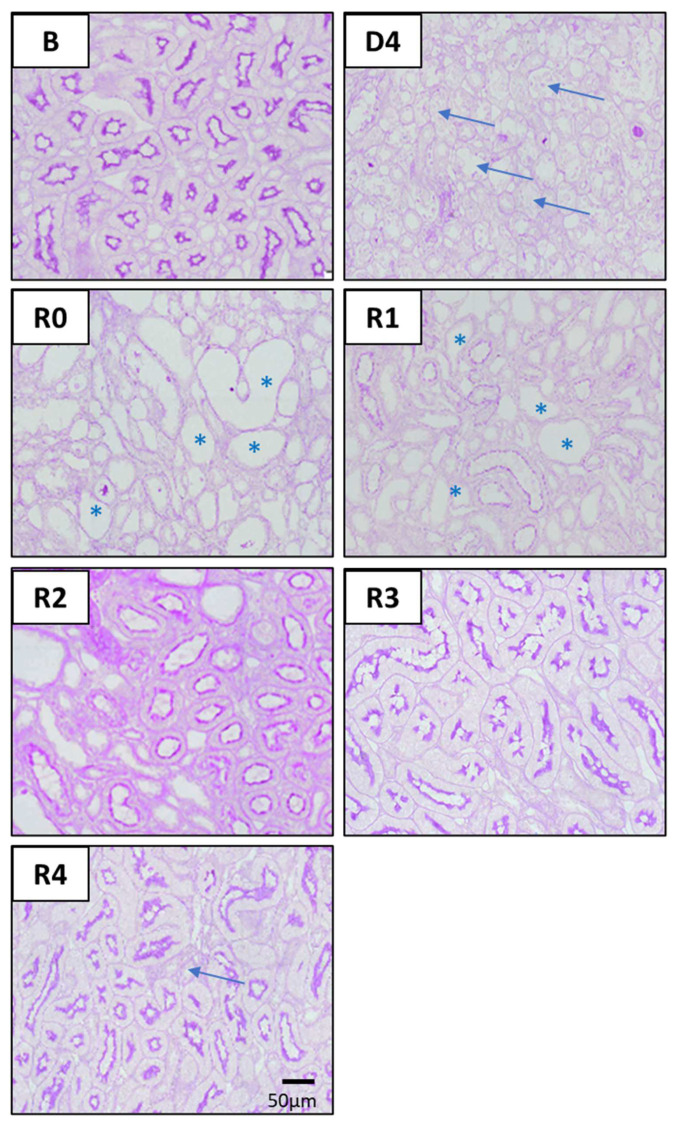
Renal histology. Representative images of kidney specimens stained with periodic acid–Schiff (PAS) at different time points of AKI evolution. (**B**): basal; (**D4**): day of maximum kidney damage after cisplatin treatment; (**R0**): day of recovery; (**R1**): 1 week after recovery; (**R2**): 2 weeks after recovery; (**R3**): 3 weeks after recovery; (**R4**): 4 weeks after recovery. Arrows in D4 indicate widespread detachment of necrotic tubular epithelial cells. Asterisks in (**R0**) and (**R1**) indicate dilated proximal tubules with loss of the brush border. Arrow in R4 indicates an area with interstitial fibrosis and tubular atrophy.

**Figure 5 biomedicines-10-01106-f005:**
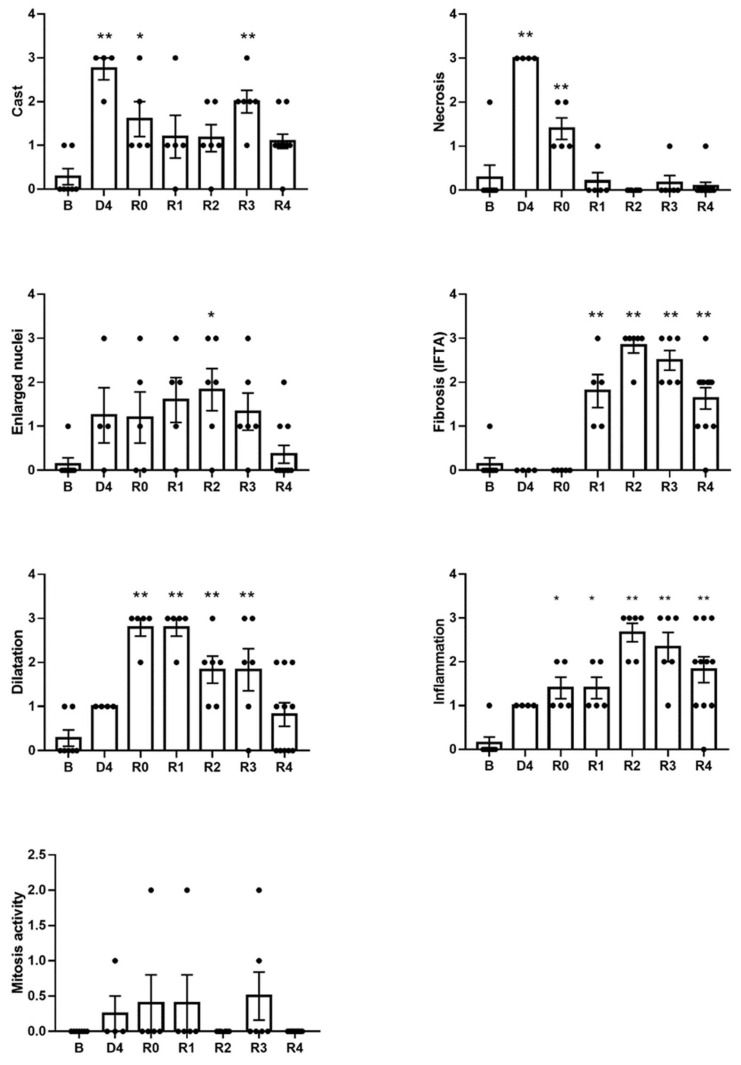
Score of several parameters of histological damage based on haematoxylin–eosin and periodic acid–Schiff staining. B: basal; D4: day of maximum kidney damage after cisplatin treatment; R0: day of recovery; R1: 1 week after recovery; R2: 2 weeks after recovery; R3: 3 weeks after recovery; R4: 4 weeks after recovery. * *p* < 0.05 vs. B; ** *p* < 0.01 vs. B.

**Figure 6 biomedicines-10-01106-f006:**
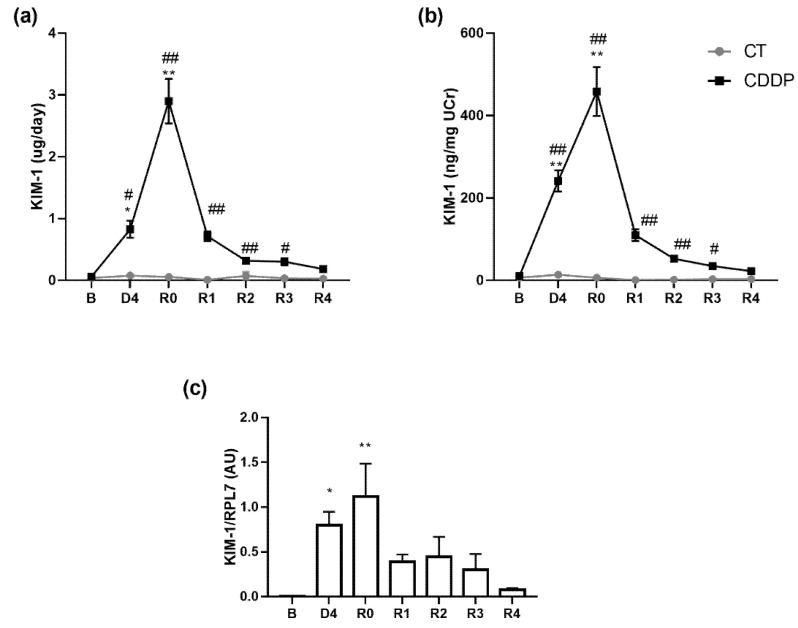
Urinary excretion of KIM-1 normalized by both urine flow (µg/day) (**a**) and urine creatinine (ng/mg UCr) (**b**) and KIM-1 mRNA expression in renal tissue (**c**). B: basal; CDDP: cisplatin treatment (5 mg·kg^−1^ body weight) group; CT: control group; D4: day of maximum kidney damage after cisplatin treatment; R0: day of recovery; R1: 1 week after recovery; R2: 2 weeks after recovery; R3: 3 weeks after recovery; R4: 4 weeks after recovery. * *p* < 0.05 vs. B; # *p* < 0.05 vs. CT; ** *p* < 0.01 vs. B; ## *p* < 0.01 vs. CT.

**Figure 7 biomedicines-10-01106-f007:**
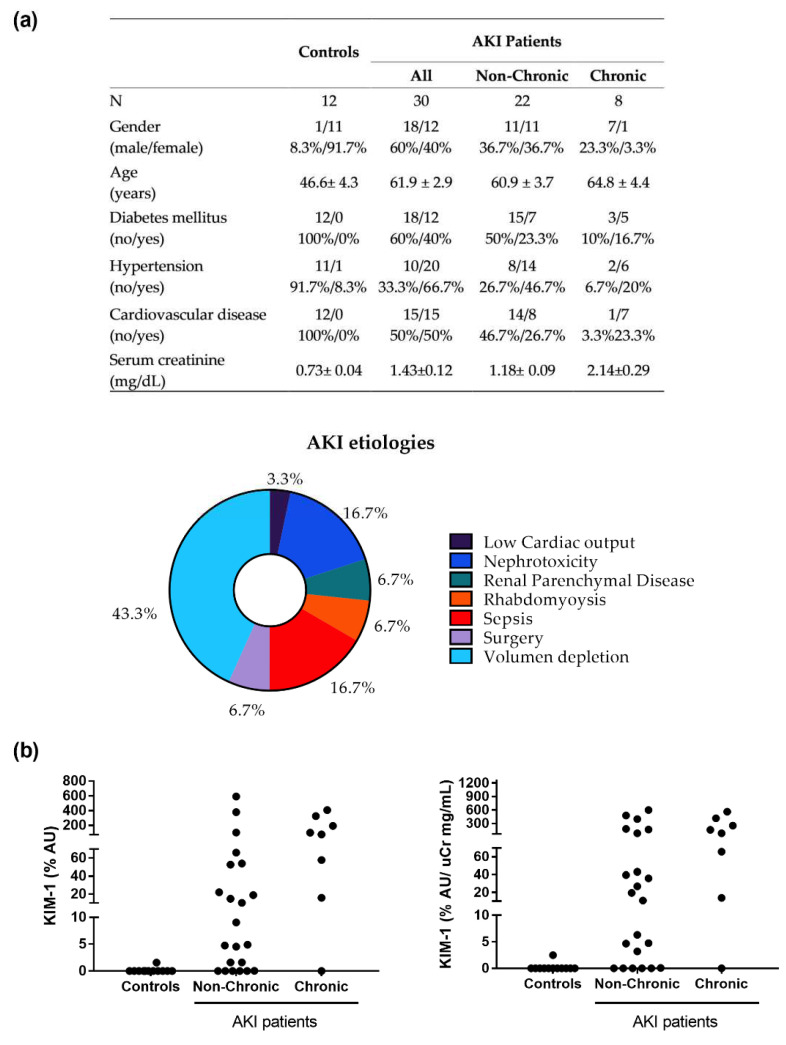
Urinary excretion of KIM-1 in patients who had recovered from an episode of AKI and controls. (**a**) Study population characteristics. (**b**) Urinary KIM-1 level after serum Cr had returned to the values prior to the AKI episode in patients with previous CKD (i.e., chronic) and in patients without previous CKD (i.e., non-chronic), and controls (**left panel**). Urinary KIM-1 levels (as in B) normalized by urinary creatinine concentration (uCr) (**right panel**). AU: arbitrary units.

**Table 1 biomedicines-10-01106-t001:** Spearman’s correlation between urinary levels of KIM-1 in recuperation points (R0–R4) and the degree of histological renal damage parameters. In bold, statistically significant differences. *p*-value in parentheses.

	KIM1 (µg/Day)	KIM1 (ng/mg CrU)
Dilatation	**0.4957 (0.001)**	**0.6286 (<0.0001)**
Cast	0.2209 (0.1651)	**0.4809 (0.0022)**
Enlarged nuclei	**0.3694 (0.0175)**	**0.4750 (0.0026)**
Mitosis activity	0.2678 (0.0993)	0.3094 (0.0587)
Inflammation	−0.0904 (0.5741)	−0.0330 (0.8441)
Necrosis	**0.5282 (0.0004)**	**0.6403 (<0.0001)**
Fibrosis (IFTA)	−0.3031 (0.054)	−0.2099 (0.206)

## Data Availability

The data presented in this study are available on request from the corresponding author.

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
