# Peer review of "Urinary KIM-1 Correlates with the Subclinical Sequelae of Tubular Damage Persisting after the Apparent Functional Recovery from Intrinsic Acute Kidney Injury"

_biomedicines, 2022, doi:10.3390/biomedicines10051106_

Round 1

Reviewer 1 Report

The study is well designed and the proposed protocol is sound and so the results can be extrapolated to the bedside. Overall, the article can be published in its current form. 

Author Response

We thank the reviewer for his/her positive evaluation of our manuscript

Reviewer 2 Report

The authors reported an evaluation of the KIM-1 role in AKI and CKD. Some considerations limit the generalizability of this paper and should be discussed.

  • As the authors reported, KIM-1 documentation in cisplatin-induced renal damage is relatively well established, with only a limited novelty derived from the documentation of increased Havcr1 transcription. Already the recovery in mice was almost complete at four weeks, so the bench to bedside approach needed an evaluation of KIM-1 excretion in mice after a long period (i.e., two-three months) to suggest a correlation to data observed in CKD patients
  • The documentation of elevated KIM-1 in CKD patients, at this time, is empirical, suffer from multiple etiologies of renal damage, and at least needs some clarification (e.g., data of clinical recovery and time of sample collection after the acute episode; eGFR and proteinuria - if possible, with discrimation between glomerular and tubular urinary proteins-)

Author Response

The authors reported an evaluation of the KIM-1 role in AKI and CKD. Some considerations limit the generalizability of this paper and should be discussed.

Authors: First, we thank the Reviewer for their comments, which have prompted us to further think on the potential implications of our study. Actually, the main focus of our study is placed on the post-AKI stage, not so much on AKI or CKD. We aimed at identifying non-invasive probes of the potential, subclinical sequelae of AKI, which would help follow renal recovery, or absence thereof.

As the authors reported, KIM-1 documentation in cisplatin-induced renal damage is relatively well established, with only a limited novelty derived from the documentation of increased Havcr1 transcription. Already the recovery in mice was almost complete at four weeks, so the bench to bedside approach needed an evaluation of KIM-1 excretion in mice after a long period (i.e., two-three months) to suggest a correlation to data observed in CKD patients.

Authors: As commented above, the aim of this study was placed on the non-invasive follow-up of subclinical sequelae of AKI, not on progression to CKD. In the Discussion, and for the sake of analyzing some perspectives, results are contextualized for their application in the identification of patients with incomplete recovery from AKI, who might be at higher risk of renal complications in the short (i.e., a new AKI episode) and long term (i.e., progression to CKD).

The documentation of elevated KIM-1 in CKD patients, at this time, is empirical, suffer from multiple etiologies of renal damage, and at least needs some clarification (e.g., data of clinical recovery and time of sample collection after the acute episode; eGFR and proteinuria - if possible, with discrimation between glomerular and tubular urinary proteins-).

Authors: The Reviewer’s comment emphasizes a very relevant issue that is, at least to some extent, addressed in the Discussion. Our purpose was not to analyze the potential pathophysiological causes of elevated KIM-1 urinary excretion in CKD patients (which is impossible with this sample size), but to show their case differentiated from non-CKD patients. This is because, in our view, it is reasonable to suspect that CKD patients might contaminate our argument of elevated KIM-1 after AKI as a sign of subclinical sequelae, as CKD patients might have elevated urinary KIM-1 before their AKI episode. If so, they would retain high levels of urinary KIM-1 (associated to the chronic process) regardless of the degree of renal recovery from AKI. In those circumstances, when unaware of their previous KIM-1 level (as in our case), it would be difficult to discern whether the elevated post-AKI KIM-1 levels are due to sequelae from AKI, to the chronic process or to a mixture of both. We show their result as they confirm our argument. As shown in Figure 7, all CKD patients (except one) have elevated levels of urinary KIM-1 when serum creatinine returned to the value patients showed before the AKI episode. Following the Reviewer’s comment, we realized it was not clearly explained, and it has now been made explicit in Material and Methods. In this context, we are not sure about the utility of the eGFR and proteinuria data. 

Reviewer 3 Report

The manuscript entitled: “Urinary KIM-1 correlates with the subclinical sequelae of tubular damage persisting after the apparent functional recovery from intrinsic acute kidney injury” investigates the role of KIM-1 as a biomarker for non-invasive follow-up of renal repair after AKI. The manuscript is well-written and interesting for the research community. Some aspects should be clarified before acceptance.

  1. In the material and methods please indicate the age of the animals used
  2. Indicate the doses of i.p pentobarbital used for anesthesia
  3. A flowchart with the study design should be included in the material and methods part
  4. In the discussion section add the strengths and limitations of the current study

Author Response

The manuscript entitled: “Urinary KIM-1 correlates with the subclinical sequelae of tubular damage persisting after the apparent functional recovery from intrinsic acute kidney injury” investigates the role of KIM-1 as a biomarker for non-invasive follow-up of renal repair after AKI. The manuscript is well-written and interesting for the research community. Some aspects should be clarified before acceptance.

Authors: We sincerely thank the Reviewer for their positive consideration of our article and for their comments to improve it.

In the material and methods please indicate the age of the animals used.

Authors: Following the Reviewer’s comment, the age of rats is now indicated.

Indicate the doses of i.p pentobarbital used for anesthesia.

Authors: Yes, the dose of pentobarbital has also been included. Thank you.

A flowchart with the study design should be included in the material and methods part.

Authors: A flowchart showing the study design is now included as Figure 1.

In the discussion section add the strengths and limitations of the current study.

Authors: We have added a sentence about the study limitations in the last paragraph of the Discussion. In our view, the strengths were already described in this same paragraph of summary and perspectives. Should the Reviewer have additional suggestions for the strengths and limitations, we stay open to including them.

Round 2

Reviewer 2 Report

The authors partially addressed all requested questions, and the paper's clarity has been improved.

Reviewer 3 Report

The authors addressed all my comments. The manuscript is ready for acceptance.